# Morphology and Molecular Phylogeny of Four Anaerobic Ciliates (Protista, Ciliophora, Armophorea), with Report of a New Species and a Unique Arrangement Pattern of Dikinetids in Family Metopidae

**DOI:** 10.3390/microorganisms13020240

**Published:** 2025-01-23

**Authors:** Song Li, Wenbao Zhuang, Xiaochen Feng, Alan Warren, Jun Gong

**Affiliations:** 1Laboratory of Microbial Ecology and Matter Cycles, School of Marine Sciences, Sun Yat-Sen University, Zhuhai 519082, China; lisong53@mail.sysu.edu.cn; 2Key Laboratory of Mariculture, Ministry of Education, College of Fisheries, Ocean University of China, Qingdao 266003, China; zhuang824@163.com (W.Z.); fxc7090@stu.ouc.edu.cn (X.F.); 3Institute of Evolution and Marine Biodiversity, Ocean University of China, Qingdao 266003, China; 4Department of Life Sciences, Natural History of Museum, London SW7 5BD, UK; a.warren@nhm.ac.uk

**Keywords:** anaerobe, diversity, Metopidae ciliate, new species, ribosomal RNA gene

## Abstract

The diversity of anaerobic ciliates is greatly underestimated owing to the limitation in sampling and cultivation when compared with their aerobic counterparts. In this study, four anaerobic ciliates, viz. *Brachonella abnormalis* sp. nov., *Brachonella contorta* (Levander, 1894) Jankowski, 1964, *Metopus contortus* (Quennerstedt, 1867) Kahl, 1932, and *Metopus major* Kahl, 1932, were investigated by live observation, protargol staining and 18S rRNA gene sequencing. *B. abnormalis* sp. nov. can be separated from its congeners by a combination of the following features: bullet-shaped cell with a life size of about 130–190 × 90–120 μm, dikinetids distributed along dorsal dome kineties, highly developed adoral zone comprised of 87–107 polykinetids, making about 450° spiralization around the long axis. The present work demonstrates that two known species, *M. contortus* and *M. major*, have a special trait never previously reported, viz. short, regularly arranged preoral dome dikinetids. Species with short, regularly arranged dome dikinetids appear in divergent clades in SSU rRNA gene trees, which may infer that this trait evolved several times. Phylogenetic analyses based on SSU rRNA gene sequence data also support the monophyly of the genus *Brachonella* and the paraphyly of the order Metopida, respectively.

## 1. Introduction

Ciliates (phylum Ciliophora) are a morphologically diverse protozoan group with body lengths generally ranging from 10 to about 2000 μm, constituting a large assemblage of unicellular eukaryotes that are distinguished by the presence of cilia and nuclear dimorphism. These protists are significant grazers of bacterial, picophytoplankton, and ultraphytoplankton production, playing significant roles in microbial food webs [1,2,3]. They are widely distributed in both aerobic and anaerobic habitats across freshwater, soil, and marine ecosystems [4,5].

Anaerobic ciliates have been useful model organisms in understanding the cellular and biochemical mechanisms facilitating eukaryotic adaptation to low-oxygen environments, such as the findings of progressively reduced and modified mitochondrion-related organelles, in conjunction with archaeal and/or bacterial symbionts that facilitate their fermentative metabolism [6,7,8]. Despite these advances, the diversity, taxonomy, and evolution of anaerobic ciliates are still needed to lay a good base for a better understanding of their systematics, ecology, and evolution [9,10].

Most free-living anaerobic ciliates belong either to class Plagiopylea or Armophorea [11,12,13,14]. The designation of Armophorea was based mainly on small subunit ribosomal RNA gene (SSU rDNA) trees; this class is thus considered a “riboclass” due to the absence of any obvious morphological synapomorphies [4]. Currently, Armophorea comprises the orders Metopida, Armophorida, and Clevelandellida. Members of the former two are mostly free-living anaerobes, while the latter consists of endosymbionts. Metopida, as one of the representative groups of Armophorea, has recently been investigated using an integrative approach that combines traditional and modern methods [15,16,17,18]. Within this order, *Metopus* and *Brachonella* are two relatively speciose genera; however, molecular data are lacking for about half of their nominal species. New species are continually discovered despite the passage of nearly 250 years since Müller’s study of *Metopus es* (Müller, 1776) Lauterborn, 1916. *Brachonella* was established by Jankowski in 1964 and includes 10 nominal species so far, although combined morphological and molecular data are available only for three of these, namely *B. contorta* (Levander, 1894) Jankowski, 1964, *B. pulchra* (Kahl, 1927) Bourland et al., 2017, and *B. comma* Bourland et al., 2022 [5,19].

To reveal the biodiversity of *Brachonella* and *Metopus* and determine the systematics of these two genera, we conducted a faunistic study in multiple provinces across China. In the present work, we investigated the morphology and ciliary pattern of a new species, *Brachonella abnormalis* sp. nov., as well as four populations assigned to three known species, viz. *B. contorta*, *Metopus contortus* (Quennerstedt, 1867) Kahl, 1932, and *M. major* Kahl, 1932. We also obtained 18S rRNA gene sequences of these five isolates and performed phylogenetic analyses.

## 2. Materials and Methods

### 2.1. Sample Collection

*Brachonella abnormalis* sp. nov. and Qingdao population of *B. contorta* were isolated from sediments of a stagnant freshwater pond with abundant reeds (water temperature 25 °C; 36°03′59.3″ N, 120°20′42.2″ E) in Qingdao, China, in May 2015 (Figure 1A,B); another population of *B. contorta* was collected from sediments of Weishan lake (water temperature 23 °C; 34°39′25.3″ N, 117°12′13.1″ E) in Jining, China, in June 2022; *Metopus contortus* was discovered in a sample of sand and sea-weed from an estuary (water temperature 12 °C, salinity 28‰; 22°30′18.6″ N, 113°57′15.6″ E) in Shenzhen, China, in March 2016 (Figure 1A,C). *Metopus major* was collected from a seawater pool in a mangrove forest (water temperature 15 °C, salinity 30‰; 20°03′34.4″ N, 110°20′15.9″ E) in Haikou, China, in December 2018 (Figure 1A,D).

### 2.2. Morphological Observations

Observations of living cells were conducted using bright-field and differential interference contrast illumination with light microscopes (Zeiss AXIO Imager D2, Jena, Germany). Protargol staining was then carried out to reveal the ciliature and nuclear apparatus [20]. The protargol reagent was prepared according to Pan et al. [21]. In vivo measurements were performed at a magnification of 40–1000×. Measurements and counts were performed at 1000× magnification. Drawings of specimens were made with the aid of a drawing device.

### 2.3. DNA Extraction and Gene Sequencing

One or more cells of each species were picked out with a glass micropipette and washed five times using sterile habitat water. Genomic DNA was extracted using a DNeasy Blood & Tissue Kit (Qiagen, Hilden, Germany) following the manufacturer’s instructions. The PCR amplification of the SSU rRNA gene was performed using Q5 PremixTaq with the forward primer 82F (5′-GAAACTGCGAATGGCTC-3′), 11F (5′-GCCAGTAGTSATATGCTTGTCT-3′), and the eukaryotic universal reverse primer 5.8SR (5′-CTGATATGCTTAAGTTCAGCGG-3′) [22]. The PCR products were sequenced bidirectionally in the Tsingke Biological Technology Company (Qingdao, China). The contigs were assembled by Seqman (DNAStar).

### 2.4. Phylogenetic Analyses

In total, 121 SSU rRNA sequences comprising five newly sequenced species and populations, 106 other armophoreans, and seven litostomateans and spirotricheans were used in the phylogenetic analyses. Accession numbers were provided to the left of species names in the phylogenetic tree. Three species in Karyorelictea, namely *Kentrophoros gracilis* (FJ467506), *Loxodes striatus* (AM946031) and *Geleia sinica* (JF437558), were used as the out-group taxa. 18S rRNA gene sequences used in the present phylogenetic analysis but not appearing in the tree are shown in Appendix A. Sequences were aligned using the MUSCLE algorithm with the default parameters via the EMBL-EBI website (https://www.ebi.ac.uk/Tools/msa/muscle/, accessed on 20 September 2023) [23]. Ambiguously aligned regions were excluded manually using the program BioEdit 7.0.5.2 [24]. Phylogenetic trees were constructed using the maximum-likelihood (ML) and Bayesian (BI) methods. ML analysis was performed in RA × ML-HPC2 version 8.2.12 [25] according to the GTRGAMMA model. Node support was assessed by 1000 bootstrap pseudoreplicates. BI analysis was performed using MrBayes 3.2.6 [26] on the CIPRES Science Gateway server. The GTR + I + G model was selected by MrModeltest v.2.0 [27] as the best substitution model. Markov chain Monte Carlo simulations were run with two sets of four chains for 1,000,000 generations with a sampling frequency of every 100 generations. The first 25% of trees were discarded as burn-in. The remaining trees were used to calculate posterior probabilities using a majority rule consensus.

## 3. Results

### 3.1. Brachonella abnormalis sp. nov.

#### 3.1.1. Diagnosis

The cell size is 130–190 × 90–120 μm in vivo, about 136 × 93 μm after protargol staining. The body is bullet-shaped, with a massive conical preoral dome nearly obscuring the short, bluntly truncated postoral part. Dorsoventral flattening of the preoral dome is rather variable. Dense brownish-black cytoplasmic granules invariably aggregated at the anterior end of the preoral dome. About 47 somatic kineties and 34 preoral dome kineties. Elongated, evenly distributed caudal cilia encircling the posterior end of the body. The adoral zone makes about 450° spiralization around the long axis and is composed of about 94 polykinetids. Dikinetids on the dorsal side of the dome clustered in groups of two or three along each kinety. Freshwater habitat.

#### 3.1.2. Type Material

One protargol slide (registration number: LS2015051101) with holotype (Figure 2I,J) and paratype specimens (registration numbers: LS2015051102) was deposited in the Laboratory of Protozoology, Ocean University of China. Relevant specimens were marked on the slide with black ink circles.

#### 3.1.3. Type Locality

A stagnant freshwater pond with abundant reeds (36°03′59.3″ N, 120°20′42.2″ E) at Qingdao, China.

#### 3.1.4. Etymology

The species-group name *abnormalis* is a composite of the prefix *ab* (negative) and the third-declension two-termination Latin adjective, normalis (=normal), referring to the dikinetids being short and regularly distributed along the dome kineties, which are different from other species.

#### 3.1.5. Description

The size is about 130–190 × 90–120 μm in vivo and 121−176 × 86−112 μm, on average 136 × 93 μm after protargol staining (n = 20). The body is bullet-shaped, slightly dorsoventrally flattened, and the posterior end is bluntly truncated; the length-to-width ratio, including the preoral dome, is 1.5:1 in protargol-stained specimens (Figure 2A−C and Figure 3A–C,E,F); the preoral dome is extremely large, broadly convex, and completely overhanging the adoral membranelles proximally, occupying 80–90% (mean 84%) of the body length (Figure 2I, J and Figure 3A−C); the distal dome brim joins the body on the ventral side (Figure 3A). The cell is light brown and translucent, with a prominent black spot (about 40 μm across) at the anterior pole due to aggregation of cytoplasmic granules (Figure 2A,G). The cortex is flexible, and kinetal furrows are conspicuous, about 5 μm apart. Cortical granules are colorless in vivo, globular (0.5–0.8 μm across), and densely packed interkinetally (Figure 2H and Figure 3H). The cytoplasm is hyaline and colorless. The macronucleus is about 50 μm across (Figure 2G), with irregular particles (0.5−1.0 μm in diameter) covering its surface, located in the preoral dome; the micronucleus is inconspicuous in vivo, usually adjacent to the middle portion of the macronucleus. The contractile vacuole is large and terminal (Figure 2C and Figure 3A); defecation is observed from the posterior pole. Food vacuoles are numerous, up to 15 μm in diameter (Figure 2G and Figure 3A). Locomotion by rapid swimming while rotating about the long axis.

Most somatic cilia are about 12 μm long, and perizonal stripe cilia are 15 μm long. Caudal cilia on the posterior cell margin are up to 20 μm long (Figure 2B and Figure 3A). On average, there are 34 preoral dome kineties, 19–25 of which are widely spaced and abutting to form the apical suture; other kineties are more densely spaced and crowded in right dorsolateral trough-like depressions (Figure 2F and Figure 3B,C). Somatic kineties range from 39 to 55 in number; there is a glabrous area at the posterior end of the cell where the cytoproct is located (Figure 3D). Ciliary rows are composed of dikinetids; every two to three dome dikinetids on the anterior dome part are distinctly clustered along each row (Figure 2E,J and Figure 3B,G); dikinetids in postoral kineties are evenly distributed as in other species of *Brachonella* (Figure 2I and Figure 3B,C). The perizonal stripe is invariably composed of dikinetids arranged in five unequal rows; rows 1–4 are long, almost the same length as the adoral zone proximally, and row 5 is conspicuously shortened and widely separated from others (Figure 2I,J and Figure 3B,C); all stripe rows are densely ciliated, rows 1–3 are closely spaced; dikinetids are inclined about 45° clockwise to the kinety axis. Dome kineties 1 and 2 extend along with perizonal stripe rows; dome kinety 1 is slightly shortened proximally; dome kinety 2 terminates near the posterior end of perizonal stripe rows at the dorsal part of the cell (Figure 3B,C). The adoral zone is composed of 87–107 polykinetids, occupying about 80% of the body length, parallel to the dome brim, spiraling about 450° and terminating on the ventral surface, with the proximal portion over-hung by the preoral dome margin but not covered by the buccal lip (Figure 2I,J and Figure 3A–D). The paroral membrane is about 30 μm long (Figure 3C,D).

### 3.2. Brachonella contorta (Levander, 1894) Jankowski, 1964 [28]

This species has been redescribed several times [5,28,29,30]. Since the live morphology and ciliature of the Qingdao population and Jining Population correspond well with those of previously reported populations, only some divergent features are documented here.

#### Description Based on Qingdao Population

The body size is about 60−95 × 35−70 μm in vivo, bullet-shaped, with an extremely large conical preoral dome nearly obscuring the short, bluntly truncated postoral part (Figure 4A,D−G). The cytostome is markedly displaced posteriorly. Conspicuous black cytoplasmic granules are invariably present at the anterior end of the preoral dome (Figure 4A,D−G). The macronucleus is globular and located in the anterior half of the cell (Figure 4A−C). The perizonal stripe is invariably comprised of five rows of dikinetids; rows 1–3 are slightly longer than rows 4 and 5, which are shortened proximally (Figure 4B,C). On average, there are 27 somatic kineties, including 20 preoral dome kineties; all dikinetids are evenly distributed (Figure 4C); kinetal furrows are conspicuous (Figure 4G). The paroral membrane is about 30 μm long. The adoral zone makes about 360° spiralization about the long axis (Figure 4B,C) and is comprised of 40−54 polykinetids. Caudal cilia are about 30 μm long, encircling the posterior end of the cell (Figure 4A,E,G).

### 3.3. Metopus contortus (Quennnerstedt, 1867) Kahl, 1932 [31]

#### Description Base on Qingdao and Jining Populations

The cell size is 70–100 × 25–55 μm in vivo and 76–106 × 28–51 μm after protargol staining. The body shape is oblong, distinctly twisted anteriorly, highly variable both in vivo and after fixation, ranging from distinctly sigmoidal to elongate triangular to less twisted obovate–oblong (Figure 5F); length to width ratio including preoral dome is 1.8–3.3 (Figure 5A and Figure 6A–F); the preoral dome is slightly convex, occupying about 55% of the body length when viewed ventrally, slightly wider than the mid-body (Figure 5A and Figure 6B), the posterior end is bluntly obconical. The cell is yellowish brown under low magnification, and conspicuous cytoplasmic granules aggregate at the anterior end of the cell (Figure 6A–F). The macronucleus is elongate–ellipsoidal, about 40 μm in length; the shape of the ends varies among different individuals (Figure 5E and Figure 6I). The micronucleus is globular, about 5 μm across, adjacent to the macronucleus. The contractile vacuole is terminal and obconical, about 15 μm across (Figure 5A and Figure 6C,E,F). The interkinetal cortical granules (less than 0.5 μm across) are scattered irregularly between kinety rows. Swims leisurely while rotating about the main axis.

Most somatic cilia are about 10 μm long, the perizonal stripe is 15 μm long, and caudal cilia are about 30 μm long (Figure 5A and Figure 6A,B). Somatic kineties number from 37 to 47 in number; inconspicuous glabrous area at the posterior end of the cell (Figure 5B,C); on average, 17 preoral dome kineties (Figure 5B,C). Ciliary rows are composed of dikinetids, both basal bodies ciliated in the oral portion of the cell; every two to three dome dikinetids on the anterior dome part distinctly clustered along each row (Figure 5C and Figure 6K), while other dikinetids are evenly distributed (Figure 5B,C and Figure 6G,J,K). The perizonal stripe extends about 50% of the body length and is composed of four or five longitudinal rows; its anterior end abuts the distal end of the adoral zone (Figure 5B,D and Figure 6G,H). The adoral zone is composed of 33–45 polykinetids, spiraling about 180° across the dorsal side, nearly transversely across the left side, and then descending nearly vertically, terminating on the right of the ventral side (Figure 5A,B and Figure 6J); the proximal portion is enclosed in the buccal cavity; the base of the proximal three or four polykinetids are short, rectangular, composed of about five rows of four basal bodies each; the base of the polykinetids at the mid-portion of the adoral zone are about 25 μm long (Figure 5B and Figure 6J).

### 3.4. Metopus major Kahl, 1932 [31]

#### Description Base on Haikou Population

The cell size is 150–230 × 45–90 μm in vivo and 134–228 × 44–73 μm after protargol staining. The body is narrowly oblong with a length-to-width ratio of 3:1 on average, not flattened, and distinctly twisted anteriorly (Figure 7A,C–F and Figure 8A,E,F). The preoral dome is broad, occupying about 60% of the body length, and is flat to centrally convex with a shallow concave brim (Figure 7A,C,D and Figure 8A,E,F). The cell is yellowish brown under low magnification, with cytoplasmic granules 1–3 μm in diameter, aggregated between the anterior end of the cell and the front part of the macronucleus (Figure 8A,C). The macronucleus is large, 72–111 μm in length in protargol preparations, usually elongate ellipsoidal, and the ratio of macronucleus length to cell length varies greatly (Figure 7E–G). The cortex is conspicuous, forming a distinct ca. 2 μm thick hyaline layer beneath the cell membrane in vivo (Figure 8G), finely furrowed by somatic kineties. Cortical granules were not observed in vivo. The contractile vacuole is terminal and obconical, about 15 μm across; the contractile vacuole pore is conspicuous, located adjacent to the cytoproct (Figure 7A,C and Figure 8A,E). Food vacuoles up to about 25 μm in diameter contain flagellates and bacteria (Figure 7A,G and Figure 8A). Swims leisurely with a corkscrew-like trajectory while rotating about the main axis.

Somatic ciliature is composed of dikinetids which are short, regularly arranged, clustered in groups of two to four along preoral dome kineties and in groups of two or three along postoral kineties. Both basal bodies are ciliated in the oral portion of the cell; the anterior basal body is usually barren in the post-oral region, with cilia about 15 μm long in vivo (Figure 7C and Figure 8E). On average, there are 59 meridional rows commencing close to the proximal margin of the adoral zone and extending to the posterior end of the body (Figure 7C and Figure 8E). Several elongated caudal cilia are up to 40–50 μm long (Figure 7A and Figure 8B,C). The perizonal stripe is invariably composed of five longitudinal rows forming oblique false kineties, shorter than the adoral zone proximally. Cilia of the perizonal stripe are about 20 μm long in vivo. The adoral zone is composed of 89–112 polykinetids, spiraling about 180° across the dorsal side, extending transversely across the left side, descending nearly vertically to end on the right of the ventral side, and the proximal portion is enclosed in the buccal cavity (Figure 7C and Figure 8E). The bases of the proximal three or four polykinetids are short, rectangular, and composed of about five rows of four basal bodies each; the bases of polykinetids at the mid-portion of the adoral zone are about 20 μm long.

### 3.5. SSU rDNA Sequence and Phylogenetic Analyses

Lengths and GC contents of the five newly obtained SSU rRNA gene sequences are as follows: *Brachonella abnormalis* sp. nov., 1593 bp, 43.57% GC; Qingdao population of *B. contorta*, 1585 bp, 43.91% GC; Jining population of *B. contorta*, 1653bp, 43.68% GC; *Metopus contortus*, 1558 bp, 44.54% GC; *M. major*, 1583 bp, 44.60% GC. The sequences of *M. contortus* and *B. contorta* from this work exhibited more than 99% similarities to conspecific populations. The phylogenetic tree of the Metopida/Clevelandellida lineage inferred from the 18S rRNA gene sequences is shown in Figure 9. The topologies of the phylogenetic trees from ML and BI analyses were almost identical; therefore, only the ML tree is shown with support values from both analyses. Sequences of members of the genus *Brachonella*, viz. *B. contorta* (see Appendix A for GeneBank accession numbers), *B. abnormalis* sp. nov., and *B. pulchra* (HM108621) group together with full support. *Metopus major*, *M. contortus*, and *M. parapellitus* cluster together with maximum support forming a clade that groups with a robust marine clade formed by three marine species of Metopus, viz. *M. paravestitus*, *M. vestitus*, and *M. spiculatus*, also with maximum support.

## 4. Discussion

### 4.1. Comparison of Brachonella abnormalis sp. nov. with Congeners

Within this genus, *B. contorta*, *B. pulchra*, and *B. comma* can be compared with the new species in terms of their ciliature and 18S rRNA gene sequence available [5,32,33]. *Brachonella abnormalis* sp. nov. is easily distinguished from these three species by the unique arrangement pattern of dikinetids in its preoral dome kineties (dikinetids clustered in groups of two or three vs. evenly distributed along each kinety). Additionally, *B. abnormalis* sp. nov. is larger in vivo (130–190 × 90–120 μm vs. 70−154 × 42−106 μm, 90–118 × 51–71 μm and 71–100 × 23–38 μm) and has higher numbers of adoral polykinetids (87–107 vs. 30–76, 53–63 and 30–35) and preoral dome kineties (39–55 vs. 5–26, 27–39 and 10–11) than *B. contorta*, *B. pulchra*, and *B. comma* (Table 1 and Appendix A).

Although information on other congeners is imperfect due to the limitations of experimental methods available at the time they were described, *B. abnormalis* sp. nov. can be separated from them based on the available data. *Brachonella lemani* (Dragesco, 1960) Jankowski, 1964 has a long-pointed tail (vs. absent in the new species) and an elongated sausage-shaped (vs. broadly ellipsoidal) macronucleus [34,35]. *Brachonella abnormalis* sp. nov. can be distinguished from *B. cydonia* (Kahl, 1927) Jankowski, 1964 by the shape of the posterior end of the cell and distribution of caudal cilia (broadly rounded, caudal cilia dispersed vs. narrowly tapered, caudal cilia in a discrete tuft) [35]. *Brachonella abnormalis* sp. nov. differs from *B. elongata* Jankowski, 1964 by the cell shape (bullet-shaped vs. elongated oval shape), the degree of adoral zone spiraling (more than 360° vs. about 270°), the number of somatic kineties (39–55 vs. 26–34). Both *B. fastigata* (Kahl, 1927) Jankowski, 1964 and *B. intercedens* Kahl, 1927 clearly differ from *B. abnormalis* sp. nov. by their pyriform (vs. ellipsoidal) body shape [28,31]. *B. mitriformis* Alekperov, 1984 has fewer somatic kineties (less than 35 vs. more than 39) and is smaller (less than 100 μm vs. more than 120 μm after protargol staining) and, thus, cannot be confused with the new species [35]. *B. abnormalis* sp. nov. is easily separated from *B. caenomorphides* Foissner, 1980 by the rounded (vs. narrowly tapered) posterior body end and the number of rows in the perizonal stripe (invariably 5 vs. 3–5) [36]. The new species differs from *B. pulchra* by the presence (vs. absence) of the anterior ciliary suture and the number of adoral polykinetids (more than 85 vs. less than 65).

### 4.2. Comparison of Metopus major with Related Forms

*Metopus major* was first reported by Kahl [31] as a variety of *M. contortus*. Compared to the latter, this form has a larger body size (250–300 μm in length in vivo), a more compressed preoral dome, and a straighter body. Esteban et al. [12] raised it to the level of species based on a Denmark population with a smaller body size and compared it with *M. contortus* from the same locality noting that the former is larger (113–191 μm vs. 89–165 μm) and has more adoral polykinetids (80 vs. 35–50) and somatic kineties (50–55 vs. 40) than the latter. Two subsequent re-descriptions of *M. contortus* showed that it could be larger (107–175 × 40–75 μm in vivo) and have more somatic kineties (48–59) than the Denmark population [29,37], which blurs the boundary between the two species and casts doubt on the decision whether *M. major* is a valid species (Table 2, Appendix A).

The Haikou population of *M. major* closely matches the Denmark population [12] in terms of cell size, and the size and shape of the macronucleus are similar to those of the Denmark population but differ from Kahl’s description. It is noteworthy, however, that the macronuclear sizes of metopids depicted in the illustrations drawn by Kahl are often smaller than those in subsequent redescriptions [5,17,19]. Therefore, we believe that the difference in macronuclear size between the Haikou and Denmark populations, on the one hand, and Kahl’s population, on the other, is not significant.

Based on the Haikou population of *M. major*, some clear differences exist between it and *M. contortus*. Firstly, the Haikou population expands the difference in the number of adoral polykinetids between the two species (89–112 vs. fewer than 71, usually fewer than 55). Secondly, *M. major* shows a preoral dome area with short, regularly distributed somatic dikinetids, i.e., arranged in groups of two or three along almost all somatic kineties (Figure 7C,D). Although not easy to distinguish, the Denmark population also has this uneven distribution of somatic dikinetids [12]. By contrast, in *M. contortus*, the uneven distribution of dikinetids is confined to the dome part, with each group comprising only two dikinetids in some populations [19], whereas the dikinetids in another part of the cell are evenly distributed.

Compared with other species in the genus *Metopus* and related genera, there is no doubt that *M. major* has a large body size (cell length ranging from 150 to 200 μm). Other forms with this size are all freshwater species. *Metopus rex* Vďačný and Foissner, 2017 has highly refractive cortical granules (vs. absent in *M. major*). In addition, the distance between the preoral dome kineties and the perizonal ciliary stripe is large enough in *M. rex* to easily separate it from *M. major* [17]. Unlike *M. major*, *M. ovalis* has conspicuously purple cortical granules and lacks elongated caudal cilia [31].

### 4.3. Comparison of Metopus contortus with Related Forms

The Shenzhen population is consistent with the variety “*Forma pellitus*” of *M. contortus* described by Kahl in terms of size and shape and is basically consistent with the re-descriptions, especially in terms of numbers of somatic kineties and adoral polykinetids, and the presence of posterior cilia (Appendix A) [12,29,30,37,38,39]. However, the populations described by Esteban [12] and Foissner [37], possess 5 perizonal ciliary stripe rows, whereas Shenzhen population possesses 4 to 5 rows and the populations described by Dragesco and Dai have 3 rows [29,39]. This species has been predominantly found in marine environments, including the Shenzhen population with the exception of the freshwater population of *M. contortus* reported by Foissner [12,29,30,37,38,39,40,41,42].

### 4.4. Uneven Distribution of Dikinetids in Dome Kineties in Brachonella abnormalis sp. nov., Metopus major and Metopus contortus

Relatively little is known about the uneven distribution of dikinetids in the dome kineties of armophorids. Although this feature was previously shown in *M. major* and *M. contortus*, it has never been reported. Consequently, its presence in other populations is difficult to determine as the feature is unrecognizable in published photomicrographs of protargol preparations [29,30]. This pattern of short, regularly distributed dikinetids was observed in the dome kineties of *Brachonella abnormalis* sp. nov. (Figure 2B,G and Figure 3E,I,J). We posit that such a pattern may be relate to adaptation to extremely muddy waters, although its function needs further exploration and that convergent evolution is responsible for the arrangement of similar patterns of kinetids in different species.

### 4.5. Phylogeny of Metopus and Brachonella Species

The present phylogenetic analysis shows that the family Metopidae is paraphyletic, which is consistent with previous studies [5,30,43]. Furthermore, expanded sampling supports the monophyly of the genus *Brachonella*. The overall topology of the present SSU rRNA gene tree corresponds well with previous analyses, the most significant difference being the position of the genus *Brachonella* [5,15,16]. With the addition of new sequences of *B*. *abnormalis* sp. nov. and *B. contorta*, the genus *Brachonella* can be clearly separated from *Metopus* by possessing a dominant peroral dome, a highly spiralized adoral zone, and the posterior location of the cytostome [5,15,28]. Nevertheless, sequences of more species of *Brachonella* are needed to determine the evolutionary relationships of this genus. The species *M*. *major*, which was collected from a seawater pool (salinity 30‰), clustered in a clade that contains other marine species of *Metopus* with high confidence supporting the assertion that habitat serves as a potential genus splitting clue [44]. The close phylogenetic relationship between *M. major* and *M. contortus* is consistent with their high morphological similarity.

As a freshwater form and the type species of this genus, *Metopus es* shows a special relationship with other non-freshwater species, although the confidence level is low (ML 58). More studies like comprehensive phylogenomic analyses and reports of more marine species are required to reveal their relationship and confirm whether marine clade represents a new genus.

## 5. Conclusions

We describe the morphology and morphogenesis of four anaerobic ciliates in this study, supporting the decision to raise the variety “*Forma major*” of M. *contortus* to the level of species. We also report a unique arrangement pattern of dikinetids that was observed in three species of the family Metopidae. Such a pattern may be an adaptation to extremely muddy waters. Phylogenetic analyses support Metopidae is paraphyletic. Notably, as a marine species, *M. major* clustered with other marine *Metopus* species in the phylogenetic tree with high support, suggesting that niche adaptation played a role in the speciation of *Metopus*. More comprehensive studies, such as phylogenomic analyses and the discovery and reporting of additional marine species, are needed to ascertain whether the marine clade within the genus *Metopus* merits recognition as a new genus.

## Figures and Tables

**Figure 1 microorganisms-13-00240-f001:**
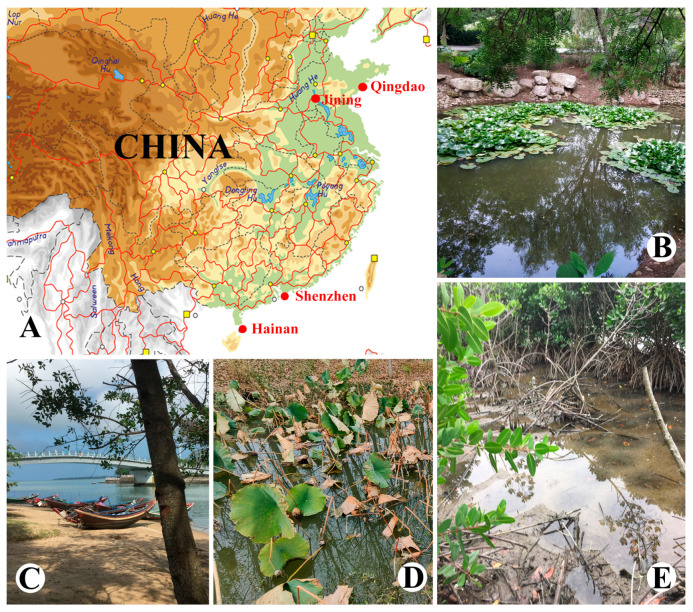
Location of sampling sites. (**A**) Partial map of China. (**B**) The stagnant freshwater pond in Qingdao where *Brachonella abnormalis* sp. nov. and *B. contorta* were isolated. (**C**) The estuary in Shenzhen where *Metopus contortus* was collected. (**D**) The lake in Jining where *B. contorta* was isolated. (**E**) The mangrove forest seawater pool in Haikou where *Metopus major* was collected.

**Figure 2 microorganisms-13-00240-f002:**
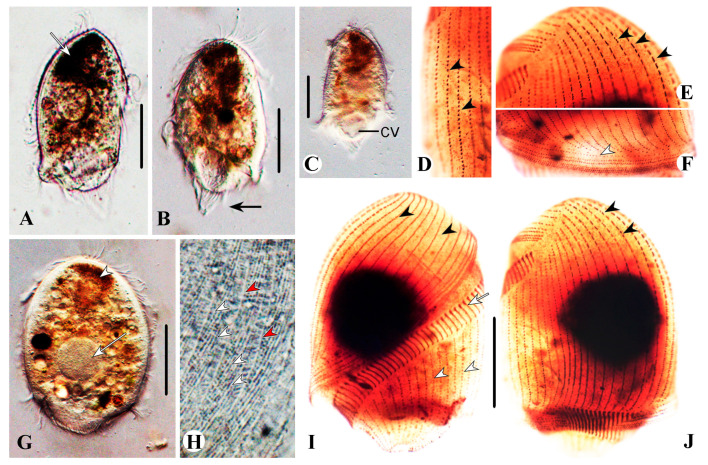
*Brachonella abnormalis* sp. nov. from life with bright-field (**A**,**H**) and differential interference contrast (**B**,**C**,**G**) microscopy and after protargol staining (**D**–**F**,**I**,**J**). (**A**) Ventral view showing dense aggregation of brown-black refractive granules in the anterior region of the cell (arrow). (**B**) Right view showing caudal cilia (arrow). (**C**) Dorsal view showing contractile vacuole. (**D**) Left part of the cell, showing evenly distributed dome dikinetids (arrowheads). (**E**) Anterior portion of the cell showing dome dikinetids (arrowheads). (**F**) Proximal margin of the preoral dome showing perizonal ciliary stripe and dome kinety 1 (arrowhead). (**G**) General view of a cell showing refractive granules (arrowhead) and macronucleus (arrow). (**H**) Interkinetal cortical granules (white arrowheads) and kinetal furrows interval (red arrowheads) in a strongly compressed cell. (**I**,**J**) Ventral (**I**) and dorsal (**J**) views of the same specimen, showing dome kineties (black arrowheads), adoral membranelles (arrow), and postoral somatic kineties (white arrowheads). CV, contractile vacuole. Scale bars: 50 µm.

**Figure 3 microorganisms-13-00240-f003:**
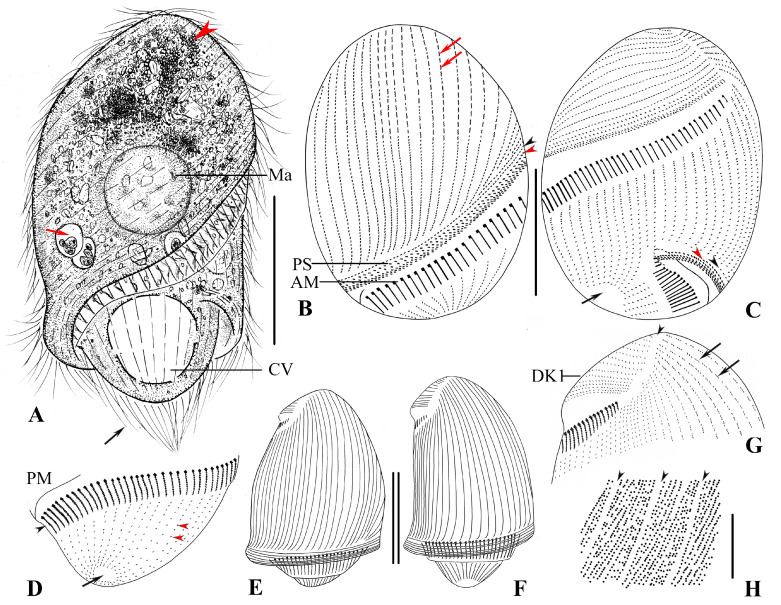
*Brachonella abnormalis* sp. nov. from life (**A**,**H**) and after protargol staining (**B**–**G**). (**A**) Ventral side of a representative specimen showing food vacuole (red arrow), aggregation of cytoplasmic granules (arrowhead), and caudal cilia (black arrow). (**B,C**) Ventral (**B**) and dorsal (**C**) views of the same specimen showing uneven arrangement of dome dikinetids (arrows in (**B**)), dome kineties 1 (black arrowhead), perizonal stripe row 5 (red arrowhead) and cytoproct area (arrow in (**C**)). (**D**) Posterior part of the cell showing adoral membranelles (black arrowhead), paroral membrane, somatic kineties (red arrowheads), and cytoproct area (arrow). (**E**,**F**) Different cell shapes in dorsal view. (**G**) Anterior part of the cell showing suture (arrowhead), dome kinety 1, and dome dikinetids (arrows). (**H**) Arrangement of cortical granules with arrowheads showing kinetal furrows interval (arrowheads). AM, adoral membranelles; CV, contractile vacuole; DK1, dome kinety 1; Ma, macronucleus; PM, paroral membrane; PS, perizonal ciliary stripe. Scale bars: 50 μm (**A**–**C**,**E**,**F**), 10 μm (**D**).

**Figure 4 microorganisms-13-00240-f004:**
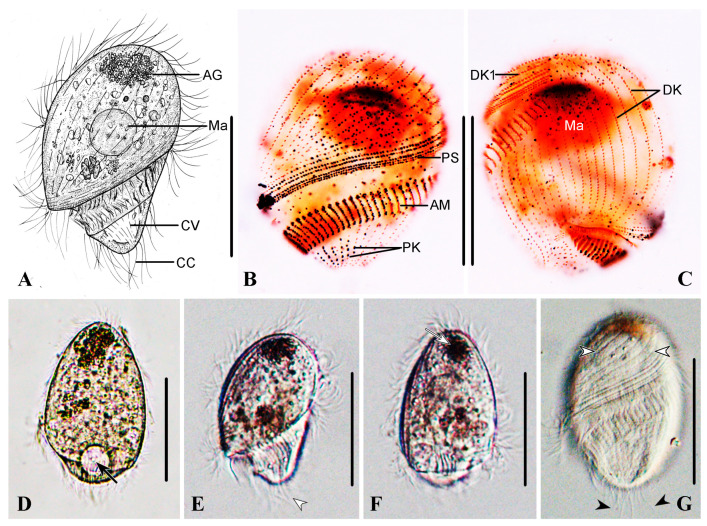
*Brachonella contorta* from life (**A**,**D**–**G**) with bright-field (**D**), differential interference contrast (**E**–**G**) microscopy, and after protargol staining (**B**,**C**). (**A**) Ventral view showing dense aggregation of brown-black refractive granules in the anterior region of the cell, contractile vacuole, and caudal cilia. (**B**,**C**) Ventral (**B**) and dorsal (**C**) views of the same specimen, showing dome kineties, adoral membranelles, and postoral kineties. (**D**) Left dorsal view showing contractile vacuole (arrow). (**E**,**F**) Ventral (**E**) and right (**F**) view of the same individual showing aggregation of granules (arrows), adoral membranelles, and caudal cilia (arrowhead). (**G**) Ventral view showing kinetal furrows interval (white arrowheads) and caudal cilia (black arrowheads). AG, aggregation of granules; AM, adoral membranelles; CC, cauda cilia; CV, contractile vacuole; DK, dome kineties; DK1, dome kinety 1; Ma, macronucleus; PK, postoral kinety; PS, perizonal ciliary stripe. Scale bars: 50 μm.

**Figure 5 microorganisms-13-00240-f005:**
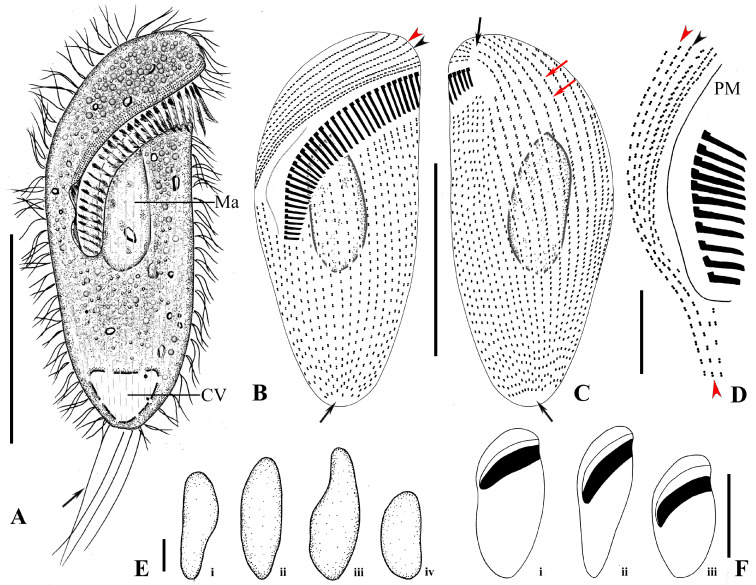
*Metopus contortus* from life (**A**) and after protargol staining (**B**–**F**). (**A**) Ventral side of a representative specimen showing general morphology, contractile vacuole, and elongated caudal cilia (arrow). (**B**,**C**) Ventral (**B**) and dorsal (**C**) views of the same specimen showing the uneven arrangement of dome dikinetids on the dorsal side (red arrows), suture (black arrow), dome kinety 1 (red arrowhead), perizonal stripe row 4 (black arrowhead) and cytoproct area (black arrows). (**D**) Detail view of the paroral membrane and perizonal ciliary stripe, showing dome kineties 1 (red arrowheads) and perizonal stripe row 4 (black arrowhead). (**E**) Shape variation of macronucleus (**i**–**iv**). (**F**) Different cell shapes in ventral view (**i**–**iii**). CV, contractile vacuole; Ma, macronucleus; PM, paroral membrane. Scale bars: 50 μm (**A**–**C**,**E**), 10 μm (**D**,**E**), 50 μm (**F**).

**Figure 6 microorganisms-13-00240-f006:**
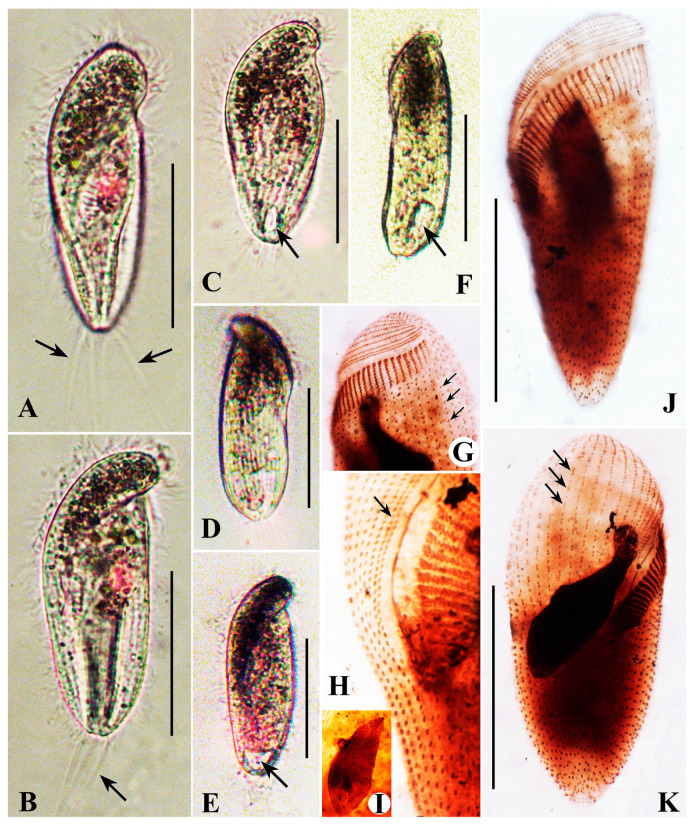
*Metopus contortus* from life with bright-field (**A**–**D**,**F**), differential interference contrast (**E**) microscopy, and after protargol staining (**G**–**K**). (**A**) Right ventral view showing aggregation of dark spherical particles, posterior cortical fold, and caudal cilia (arrows). (**B**) Right lateral view showing truncated distal end and caudal cilia (arrow). (**C**) Ventral view showing contractile vacuole (arrow). (**D**) Left view showing proximal margin of the preoral dome and contractile vacuole. (**E**) Right ventral view showing the contractile vacuole (arrow). (**F**) Left ventral view showing the contractile vacuole (arrow). (**G**) Anterior part of the cell showing the adoral membranelles and somatic kineties (arrows). (**H**) Details of perizonal ciliary stripe (arrow) and adoral membranelles. (**I**) Macronucleus surrounded by numerous irregular granules. (**J**) Left ventral view of a specimen showing adoral membranelles, perizonal ciliary stripe, and macronucleus. (**K**) Right ventral view of a specimen showing uneven arrangement of dome dikinetids (arrows). Scale bars: 50 µm.

**Figure 7 microorganisms-13-00240-f007:**
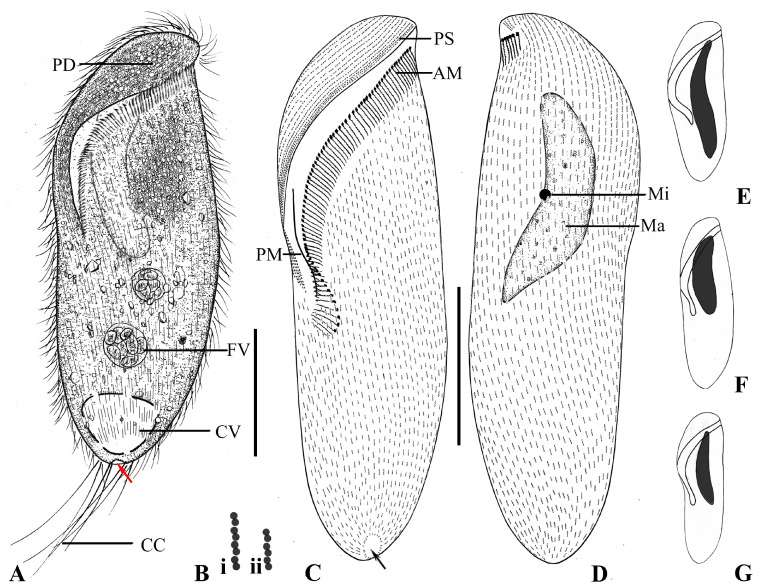
*Metopus major* from life (**A**) and after protargol staining (**B**–**G**). (**A**) Ventral side of a representative specimen showing general morphology, contractile vacuole, elongated caudal cilia, and cytoproct area (arrow). (**B**) Showing a cluster of 4 (**i**) and 3 (**ii**) dikinetids. (**C**,**D**) Ventral (**C**) and dorsal (**D**) views of the same specimen, showing the ciliature, nuclear apparatus and cytoproct area (arrows). (**E**–**G**) Showing different shapes of the macronucleus and its length compared to the body length. AM, adoral membranelles; CC, caudal cilia; CV, contractile vacuole; FV, food vacuole; Ma, macronucleus; Mi, micronucleus; PD, preoral dome; PM, paroral membrane; PS, perizonal ciliary stripe. Scale bars: 50 μm (**A**,**C**,**D**).

**Figure 8 microorganisms-13-00240-f008:**
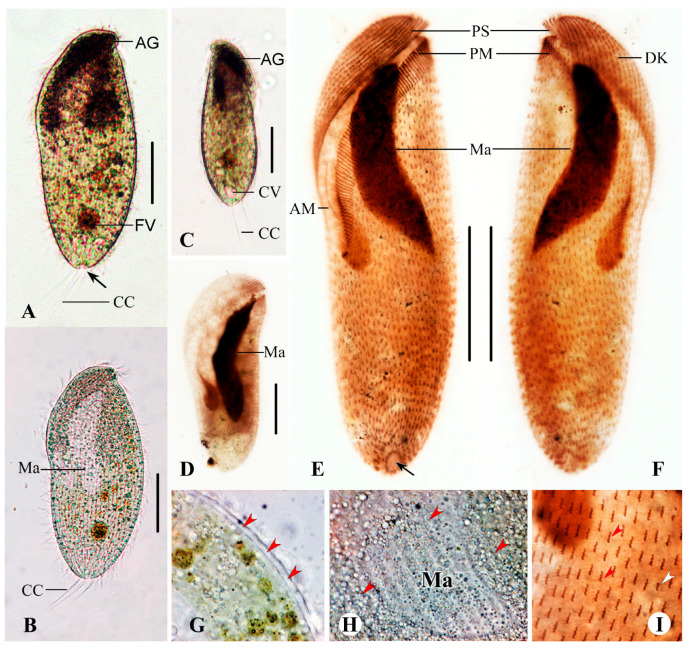
*Metopus major* from life with bright-field (**A**,**C**), differential interference contrast (**B**,**G**,**H**) microscopy, and after protargol staining (**D**–**F**,**I**). (**A**) Ventral view showing the aggregate of spherical particles, food vacuole, contractile vacuole pore (arrow), and caudal cilia. (**B**) Ventral view showing macronucleus and elongated caudal cilia trailing behind the cell during swimming. (**C**) Showing oblong outline in left ventral view. (**D**) Showing an elongated macronucleus, which is about 2/3 of the length of the cell. (**E**,**F**) Ventral (**E**) and dorsal (**F**) views of the same specimen, showing the ciliature and macronucleus. Arrow indicates the cytoproct area (arrow). (**G**) Thick hyaline cortex (arrowheads). (**H**) Dense aggregation of spherical particles (arrowheads) around macronucleus. (**I**) Showing clusters of 2 (white arrowhead) and 3 (red arrowheads) dikinetids on the dorsal side of the preoral dome. AG, aggregation of granules; AM, adoral membranelles; CC, caudal cilia; CV, contractile vacuole; DK, dome kineties; FV, food vacuole; Ma, macronucleus; PM, paroral membrane; PS, perizonal ciliary stripe. Scale bars: 70 μm (**A**–**F**).

**Figure 9 microorganisms-13-00240-f009:**
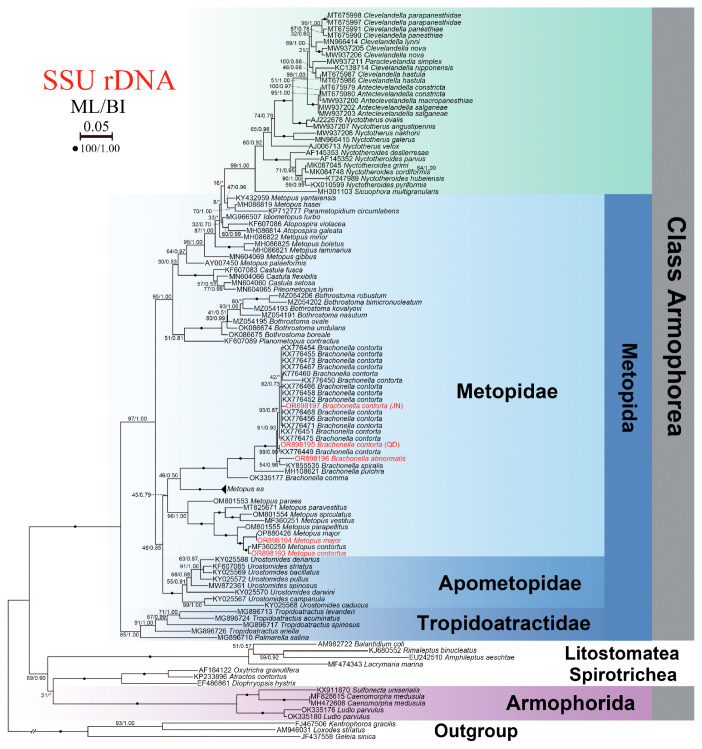
Phylogenetic tree based on SSU rDNA sequences. Numbers near branches represent BI posterior probabilities and non-parametric bootstrap values from ML. Disagreements in BI and ML tree topologies are indicated by ‘*’. All branches are drawn to scale. The scale bar corresponds to 10 substitutions per 100 nucleotide position. GenBank accession numbers are given for each species.

**Table 1 microorganisms-13-00240-t001:** Morphometric characteristics of *Brachonella abnormalis* sp. nov. (first lines), *Brachonella contorta* (Levander, 1894) Jankowski, 1964, Qingdao population (second lines), and Jining population (third lines) based on protargol-stained specimens. Measurements are in μm.

Characteristics	Mean ± SD	M	CV	Min–Max	*n*
Body, length	136.0 ± 12.3	136.1	9.1	121–176	20
	71.7 ± 10.3	74	14.4	50–88	20
	52.5 ± 4.4	52	8.4	45–62	24
Body, width	92.9 ± 7.3	92.5	7.9	86–112	20
	49.8 ± 6.7	51.5	13.4	33–60	20
	43.6 ± 4.5	45	10.3	35–54	24
Body length-to-width ratio	1.5 ± 0.1	1.5	8.5	1.3–1.7	20
	1.4 ± 0.1	1.4	5.6	1.2–1.6	20
	1.2 ± 0.1	1	5.2	1.1–1.3	24
Anterior body end to proximal end of	116.5 ± 13.2	118.6	11.3	97–144	20
adoral zone, distance	64.6 ± 8.2	65.1	12.7	51–76	20
	44.2 ± 4.2	44	9.6	37–54	24
Distance anterior body end to proximal	84.3 ± 3.1	84.8	3.6	80–90	20
end adoral zone–body length, ratio in %	84.1 ± 3.0	84	3.5	80–89	20
	84.2 ± 4.1	83.7	4.8	77–94	24
Macronucleus, length	49.7 ± 10.3	50	20.7	34–69	9
	29.1 ± 4.4	29	15.1	22–36	20
	26.8 ± 4.7	228	17.4	17–34	17
Macronucleus, width	41.0 ± 8.5	40	20.8	32–54	9
	24.3 ± 5.3	24.5	21.9	15–34	20
	20.2 ± 3.8	20	18.8	13–25	17
Adoral polykinetids, number	94.0 ± 6.8	92	7.2	87–107	9
	46.0 ± 3.7	46	8.1	40–54	19
	41.3 ± 3.8	42	9.2	33–49	26
Somatic kineties, number	47.3 ± 5.6	48	11.8	39–55	10
	26.6 ± 3.4	26	12.9	21–32	17
	37.5 ± 2.7	38	7.1	33–43	26
Preoral dome kineties, number	34.3 ± 2.2	35	6.3	29–37	10
	20.3 ± 1.7	20.5	8.1	17–23	17
	15.6 ± 1.3	16	8.4	12–18	26
Perizonal ciliary stripe rows, number	5.0 ± 0.0	5	0	5	10
	5.0 ± 0.0	5	0	5	20
	5.0 ± 0.0	5	0	5	26
Macronucleus, number	1.0 ± 0.0	1	0	1	10
	1.0 ± 0.0	1	0	1	20
	1.0 ± 0.0	1	0	1	26
Micronucleus, number	1.0 ± 0.0	1	0	1	6
	1.0 ± 0.0	1	0	1	16
	1.0 ± 0.0	1	0	1	15

Abbreviations: CV, coefficient of variation (%); M, median; Max, maximum; Min, minimum; *n*, number of specimens examined; SD, standard deviation.

**Table 2 microorganisms-13-00240-t002:** Morphometric characteristics of *Metopus contortus* Lauterborn, 1916, Shenzhen population (first lines) and *Metopus major* Kahl, 1932, Haikou population (second lines) based on protargol-stained specimens. Measurements are in μm.

Characteristics	Mean ± SD	M	CV	Min–Max	*n*
Body, length	91.0 ± 8.4	90	9.2	76–106	22
	174.6 ± 24.8	174	14.2	134–228	20
Body, width	37.6 ± 5.5	38	14.6	28–51	22
	57.3 ± 8.0	58	13.9	44–73	20
Body length-to-width ratio	2.5 ± 0.4	2.4	17	1.8–3.3	22
	3.1 ± 0.3	3.2	9.8	2.5 –3.5	20
Anterior body end to proximal end of	50.4 ± 6.1	50	12.1	37–59	22
adoral zone, distance	107.1 ± 11.4	105.5	10.7	89–132	20
Distance anterior body end to proximal	55.5 ± 5.4	55	9.7	46–65	22
end adoral zone–body length, ratio in %	61.9 ± 6.1	61.9	9.9	51–72	20
Macronucleus, length	37.5 ± 6.5	39.5	17.4	22–47	22
	87.3 ± 12.17	86.5	13.9	72–111	20
Macronucleus, width	13.1 ± 2.6	12	19.6	10–18	22
	20.3 ± 5.8	19	28.8	15–37	20
Adoral polykinetids, number	37.5 ± 3.9	36.5	10.5	33–45	16
	98.1 ± 6.6	97	6.8	89–112	19
Somatic kineties, number	41.8 ± 2.6	42	6.3	37–47	22
	58.9 ± 4.4	58	7.5	54–74	19
Preoral dome kineties, number	17.3 ± 2.4	17	13.8	12–23	22
	24.7 ± 2.0	25	8.1	22–29	19
Perizonal ciliary stripe rows, number	4.6 ± 0.5	5	10.9	5	11
	5.0 ± 0.0	5	0	5	20
Macronucleus, number	1.0 ± 0.0	1	0	1	22
	1.0 ± 0.0	1	0	1	20
Micronucleus, number	1.0 ± 0.0	1	0	1	18
	1.0 ± 0.0	1	0	1	8

Abbreviations: CV, coefficient of variation (%); M, median; Max, maximum; Min, minimum; *n*, number of specimens examined; SD, standard deviation.

## Data Availability

The data presented in the study are deposited in the GenBank database (https://www.ncbi.nlm.nih.gov/genbank, accessed on 20 September 2023), accession number: OR898193–OR898197.

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
