# Peer review of "Morphology and Molecular Phylogeny of Four Anaerobic Ciliates (Protista, Ciliophora, Armophorea), with Report of a New Species and a Unique Arrangement Pattern of Dikinetids in Family Metopidae"

_microorganisms, 2025, doi:10.3390/microorganisms13020240_

Round 1
Reviewer 1 Report
Comments and Suggestions for Authors
The study describes some new species of ciliates, which initially seems to be an interesting and novel topic, however, the manuscript is written not in a comprehensive way. The abstract lacks a clear justification of the scope of the study, as well as a generic conclusion. Similarly and more importantly, the introduction does not provide insights for the need of the study and is very short, whereas the discussion does not provide useful inferences to a brad audience. Why does it worth to investigate these ciliates? Are they pathogenic? What else do we know about them? All this info is lacking from the manuscript. While methodologically the manuscript is generally correct, as written is totally local interest oriented. The authors should have done more work to elaborate the discussion and emphasize the findings.
For instance, in line 33 [1-20] refers to too many studies that I am not sure if they are necessary. I recommend to reduce this number
The phylogenetic tree of Figure 9 is very hard to understand. It should be somehow revised to be visible
Author Response
Comments 1: The study describes some new species of ciliates, which initially seems to be an interesting and novel topic, however, the manuscript is written not in a comprehensive way. The abstract lacks a clear justification of the scope of the study, as well as a generic conclusion. Similarly and more importantly, the introduction does not provide insights for the need of the study and is very short, whereas the discussion does not provide useful inferences to a brad audience. Why does it worth to investigate these ciliates? Are they pathogenic? What else do we know about them? All this info is lacking from the manuscript. While methodologically the manuscript is generally correct, as written is totally local interest oriented. The authors should have done more work to elaborate the discussion and emphasize the findings.
Response 1: Thank you for helping us improve the manuscript. I have added more information in the Introduction section regarding the importance of anaerobic ciliates and the status of taxonoimic studies of Metopus. A conclusion section is also supplied in the revised version. All the changed sentences have been highlighted in YELLOW, see lines 32-57 and 476-490.
Comments 2: For instance, in line 33 [1-20] refers to too many studies that I am not sure if they are necessary. I recommend to reduce this number
Response 2: Many references have been deleted.
Comments 3: The phylogenetic tree of Figure 9 is very hard to understand. It should be somehow revised to be visible
Response 3: We have supplied new versions of figures of high resolution.
Reviewer 2 Report
Comments and Suggestions for Authors
The authors have analyzed four species of anaerobic ciliates in depth at the morphological and genetic levels. This information will allow specialists to study these species in more depth. Overall, the article is fairly new and can be recommended for publication after several serious shortcomings have been eliminated.
1. It is better to remove the words “Four,” “Alveolata,” “a New Species,” “sp. nov.” from the title of the article.
2. A short, simple abstract of 6-8 sentences should be added to the article, understandable for journalists and bloggers who do not understand ciliates.
3. Lines 33, 35: it is unacceptable to place more than 2-3 references in the same brackets. Readers should be familiar with (1) taxonomic diversity, (2) geographic distribution, (3) biotopes, (4) trophic relationships of the studied group of ciliates.
4. Lines 43-54 with minor changes should be moved to the abstract of the article.
5. The introduction of the article does not allow a non-specialist in this group to get an idea of ​​the relevance of the study.
6. Line 59-65: the coordinates are indicated by the authors with an accuracy of 30 meters. For ciliates, this is a very large inaccuracy. It is necessary to add tenths of a second, that is, to increase the accuracy of the sampling site to 3 meters. This will allow other authors to repeat the results of this study in tens and hundreds of years.
7. Line 72-76: it is important to show the properties of the lens (achromatic, apochromatic, planapochromatic, etc.), it is very important to use object micrometers and a digital video camera of at least 3 megapixels. The parameters indicated by the authors do not allow me to judge the reliability of the established morphological differences.
8. Figures 3, 4, 6 and others: I recommend using a standard-length bar: 5, 10, 50, 100, 200 or 500 micrometers. The authors allow themselves to use an arbitrary bar length, which indicates that the research is inaccurate.
9. The figures are of poor quality: for example, Figure 5 is so reduced in size (use the *.bmp or *.tiff format without compression, not *.jpg with high compression) that individual pixels are visible when increasing its scale to 400%. This is unacceptable for taxonomic articles.
10. Figure 9 does not allow you to read the text on it. How to review such a manuscript?
11. Table 1: in the first column after the name of the characteristic, you must indicate the unit of measurement - micrometers, separated by a comma. A second column must be added to the table and the species and population names must be written in it. It is necessary to conduct a correct multiple comparison of samples (for example, the Tukey test). It is necessary to delete the column "M" and "CV". It is necessary to combine the data "Mean +- SD", "Min - Max" into one column. The numbers in the column "Mean +- SD" should be marked with a superscript letter according to the results of the Tukey test (comparison of the authors' data with data on already described species). Rounding of numbers in the table is careless. For each characteristic, the authors should choose some rounding of numbers (either to tenths or to hundredths) and adhere to it in all 4 rows of the table.
12. Readers want to see the differences between the species in Table 1 and closely related species described earlier. This information is missing in the article. Such a table should be placed in the Discussion.
13. The Conclusion section is highly desirable in similar publications. It must be added.
Author Response
Comments 1: It is better to remove the words “Four,” “Alveolata,” “a New Species,” “sp. nov.” from the title of the article.
Response 1: Agreed. Now the title of the article has been modified as: "Morphology and Molecular Phylogeny of Four Anaerobic Ciliates (Protista, Ciliophora, Armophorea), with Report of a new Species and a unique arrangement pattern of dikinetids in Family Metopidae".
Comments 2: A short, simple abstract of 6-8 sentences should be added to the article, understandable for journalists and bloggers who do not understand ciliates.
Response 2: Thank you for pointing this out. I have added more information in Introduction, so that readers could understand ciliates better. These parts have been highlighted in YELLOW (see lines 32-50).
Comments 3: Lines 33, 35: it is unacceptable to place more than 2-3 references in the same brackets. Readers should be familiar with (1) taxonomic diversity, (2) geographic distribution, (3) biotopes, (4) trophic relationships of the studied group of ciliates.
Response 3: Agreed. I retained the 3 most relevant references here. Meanwhile, relevant content like taxonomic diversity and geographic distribution has been added to the part mentioned in Response 2.
Comments 4: Lines 43-54 with minor changes should be moved to the abstract of the article.
Response 4: Based on this comment, we have abbreviated this part. The changed part can be found in lines 67-70.
Comments 5: The introduction of the article does not allow a non-specialist in this group to get an idea of ​​the relevance of the study.
Response 5: The modification was made at Line 51-57 in response to this problem. We added more information about this group.
Comments 6: Line 59-65: the coordinates are indicated by the authors with an accuracy of 30 meters. For ciliates, this is a very large inaccuracy. It is necessary to add tenths of a second, that is, to increase the accuracy of the sampling site to 3 meters. This will allow other authors to repeat the results of this study in tens and hundreds of years.
Response 6: Thank you for pointing this out. We relocated the sampling sites and provided more precise coordinates according to your suggestion
Comments 7: Line 72-76: it is important to show the properties of the lens (achromatic, apochromatic, planapochromatic, etc.), it is very important to use object micrometers and a digital video camera of at least 3 megapixels. The parameters indicated by the authors do not allow me to judge the reliability of the established morphological differences.
Response 7: We have provided the information about the microscope in the methodology section. We confirm that our microscope and camera can meet the requirements. However, the pictures we provided have caused misunderstandings due to excessive compression.
Comments 8: Figures 3, 4, 6 and others: I recommend using a standard-length bar: 5, 10, 50, 100, 200 or 500 micrometers. The authors allow themselves to use an arbitrary bar length, which indicates that the research is inaccurate.
Response 8: Agree. We were used to controlling the length of the scale bar to be around one-third of the body length of the ciliate before, but your suggestion is more reasonable. We have replaced them with the standard-length bars.
Comments 9: The figures are of poor quality: for example, Figure 5 is so reduced in size (use the *.bmp or *.tiff format without compression, not *.jpg with high compression) that individual pixels are visible when increasing its scale to 400%. This is unacceptable for taxonomic articles.
Response 9: We have changed all figures. The reason for the low quality of the pictures is that I over-compressed the original figures.
Comments 10: Figure 9 does not allow you to read the text on it. How to review such a manuscript?
Response 10: I have replaced it with a high quality edition.
Comments 11: Table 1: in the first column after the name of the characteristic, you must indicate the unit of measurement - micrometers, separated by a comma. A second column must be added to the table and the species and population names must be written in it. It is necessary to conduct a correct multiple comparison of samples (for example, the Tukey test). It is necessary to delete the column "M" and "CV". It is necessary to combine the data "Mean +- SD", "Min - Max" into one column. The numbers in the column "Mean +- SD" should be marked with a superscript letter according to the results of the Tukey test (comparison of the authors' data with data on already described species). Rounding of numbers in the table is careless. For each characteristic, the authors should choose some rounding of numbers (either to tenths or to hundredths) and adhere to it in all 4 rows of the table.
Response 11: Thank you for your comment, I have already revised Table 1. Nevertheless, the Tukey test was not carried out. The primary reason is, many published species, especially belonging to the taxa lacking morphological studies, such as the two genera involved in this paper, date back to a rather remote past and comprehensive data are lacking. In this case, merely an average value or a range is provided. Additionally, obvious differences exist in the data among different populations of a multitude of species. Generally, in ciliate morphological research, the prevalently adopted practice is to ascertain whether there is an interval between the ranges of two species, that is, to verify whether the minimum value of one species exceeds the maximum value of another.
Comments 12: Readers want to see the differences between the species in Table 1 and closely related species described earlier. This information is missing in the article. Such a table should be placed in the Discussion.
Response 12: Thank you for your comment. In the discussion section of this version, I mainly focus on the two species, Brachonella abnormalis and Metopus major, and have carried out the comparison and discussion among the closely related species within the genus, including Brachonella contorta and Metopus contortus with the data from Table 1. I avoid excessive mention of B. contorta and M. contortus, this is because, as I have pointed out in the results section, these two species especially B. contorta have already been described and compared with related species in other articles.
The reasons why these two species are re-described in this article are not only to provide new population descriptions with 18S rDNA sequences but also because these two species are the closest to B. abnormalis and M. major in morphology and have similar habitats. Meanwhile, M. contortus also has the feature of the arrangement pattern of dikinetids described in the article. I add more information and a table (Table S2) about these two species. The changed parts are marked in Yellow in the Line 433-442.
Comments 13: The Conclusion section is highly desirable in similar publications. It must be added.
Response 13: Thank you for pointing this out. I have added a conclusion in the end. The added parts are marked in red in the Line 476-490.
Round 2
Reviewer 1 Report
Comments and Suggestions for Authors
The authors followed my comments and the revised version is definitely improved
Author Response
Comment:The authors followed my comments and the revised version is definitely improved.
Response:Thank you for your comments.
Reviewer 2 Report
Comments and Suggestions for Authors
In Table 1, it will be difficult for readers to navigate among the many lines, which of them refers to which species. Since the authors do not want to compare species, I recommend grouping the table by species: A continuous line - the Latin name of the species, then 10 lines about this species, another continuous line - the second Latin name of the species, again 10 lines of its morphology. And so on. The rest of the article can be printed.
Author Response
Comment: In Table 1, it will be difficult for readers to navigate among the many lines, which of them refers to which species. Since the authors do not want to compare species, I recommend grouping the table by species: A continuous line - the Latin name of the species, then 10 lines about this species, another continuous line - the second Latin name of the species, again 10 lines of its morphology. And so on. The rest of the article can be printed.
Response:Thank you for your comment. This table is indeed too large. I tend to divide this table into two tables by genus. This reduces the size of each table and facilitates the comparison of species within the same genus as involved in the discussion. Meanwhile, I have also made modifications to the details of the tables to enhance their readability.